# 2D Optimal Trajectory Planning Problem in Threat Environment for UUV with Non-Uniform Radiation Pattern

**DOI:** 10.3390/s21020396

**Published:** 2021-01-08

**Authors:** Andrey A. Galyaev, Pavel V. Lysenko, Victor P. Yakhno

**Affiliations:** Institute of Control Sciences of RAS, 117997 Moscow, Russia; pashlys@yandex.ru (P.V.L.); vic_iakhno@mail.ru (V.P.Y.)

**Keywords:** UUV path/trajectory planning, non-detection probability, non-uniform radiation pattern

## Abstract

Path planning is necessary in many applications using unmanned underwater vehicles (UUVs). The main class of tasks is the planning of safe routes with minimal energy costs and/or minimal levels of emitted physical and information signals. Since the action planner is on board the UUV, the main focus is on methods and algorithms that allow it to build reference trajectories while minimizing the number of calculations. The study is devoted to the problem of the optimal route planning for a UUV with a non-uniform radiation pattern. The problem is stated in the form of two point variational problem for which necessary and sufficient optimality conditions are proved. Particular attention is paid to cases where optimality conditions are not met. These cases are directly related to found specific forms of a radiation pattern. Sufficient optimality conditions are extended on the class of two-link and multi-link motion paths. Software tools have been developed and computer simulations have been performed for various types of radiation patterns.

## 1. Introduction

Various military and civil engineering applications that deal with the search of the optimal trajectories for space, air, naval and ground vehicles cover control tasks with various target functions, resource and control constraints. Examples of such tasks are: minimizing the risk of detection by aircraft radars; minimizing the risk of detection of a submarine by various sensors through different physical fields; minimizing cumulative damage during the passage of contaminated areas, etc. Despite the large variety of controlled mobile vehicles, the control tasks associated with them have common traits. First of all, these are traditional mission planning tasks, which include setting a safe course taking into account natural and artificial constraints (terrain, hydrology, weather conditions), as well as maintaining stability on a given course. In addition, the problem of optimal maneuvering in emergency situations (sharply changing weather conditions, etc.), as a rule, is typical. The success of the mission is estimated by the value of a certain function (optimization criterion), the minimization of which is the main task of the control system. Classical optimization criteria are associated with minimizing the energy costs, time (velocity problem), or miss criterion. Recently, there has been interest in non-traditional criteria, such as increasing the covertness of movement (when moving in a threat environment, considering the map of potential threats). It has led to a new area of control problems known as “trajectory/path planning in the threat environment.” The threat environment is defined as a set of agents, called conflicting, that the controlled vehicle must avoid while performing its main task [1,2,3,4,5,6].

The route planning problems are usually associated with path planning (PP) and trajectory planning (TP) problems. The path planning task consists of determining the points through which the UUV must pass to reach the prescribed destination from the starting position, while describing the UUV’s movement over time is considered to be the trajectory planning task [7]. The approach describing detection mechanisms given in [8,9,10] is used to formulate trajectory planning problem for UUV evading from detection. The estimation of the integral level of the signal that is sent to the input of a spatially distributed information and an observation system during the entire observation time is performed. In literature such a system is called a sensor [10]. The estimation of the signal integral level at the sensor input is fulfilled to solve the TP-problem with optimizing the control law for a mobile vehicle that moves from a fixed starting point to a fixed end point of the route during a given period of time. The UUV’s goal is an evasion from the detection by an observer (group of observers) located in a given area. This approach corresponds to the process of mobile vehicle detection based on an estimation of primary emitted physical signals [6]. Therefore statements of such problems are based on the estimation of physical signals emitted by the vehicle. The formalization of these problems can differ in dependence of many parameters: physical nature of detection fields, classes of acceptable control, the type of quality criteria, the number of detectors, the volume of information available to conflicting parties. As a result, based on known data, a distribution map of normalized or absolute levels of risks (threats) is created, as shown in Reference [11].

Similar approaches are used in Reference [2], where the problem of the automated path planning of combat unmanned aerial vehicles (UAVs) in the presence of radar-controlled surface-to-air missiles is considered and solved. A pre-built map of aircraft damage risks is based on the interaction of three subsystems: the aircraft and its characteristics, the radar and its capabilities, and the missile and its striking properties. Based on this map, the route with the lowest risk of damage is found.

As mentioned above, on the one hand, there is a variety of mathematical criteria describing the success of missions performed by mobile vehicles: the probability of rescue, the motion time on the trajectory, the mathematical expectation of the time interval until the first detection on the path, the path length [4,8,9,10,12,13,14]. On the other hand, despite this variety there are too few analytical results available to realize optimal trajectories in TP problems. In Reference [5], the optimal path consists of lines and circular arcs (2D Dubins curves). In Reference [10] the optimal path is presented by circle or Legandre’s functions arcs as well as in [15]. Indeed, numerical algorithms are well described in the literature.

At the initial stage, the mission planner is faced with one of two tasks:•minimizing the integral risk for a given trajectory length;•minimizing the length of the UUV motion path for a given value of the integral risk.

These problems are formulated as two-point problems of variations calculus or as optimal control problems on the plane in the presence of coordinate, phase, and integral constraints, and can be solved by one of the standard numerical methods that will set the trajectory of the UUV as the projection on the horizontal plane from 3D-space. Planning the UUV trajectory in the projection on a vertical plane is carried out using, for example, a terrain map, on which the mission planner interactively chooses the depth-levels of the trajectory, setting reference points for the beginning and end of the movement on the specified depth. Horizontal movement on a given depth is carried out along a flat trajectory obtained from solving the variational problems discussed above. The smooth conjugation of the trajectories of adjacent depths can be performed by a polynomial or spline approximation, taking into account the curvature constraint. An example of on-board realization of mission-planner for a mobile underwater robot is described in [16].

A good summary of path planning algorithms for UUVs is given in Reference [7]. These algorithms include the shortest graph-based path algorithm known as *A**, the artificial potential field algorithm, sequential quadratic programming and so on. Additionally, an approach based on ant-colony behavior is often used [11,17]. In recent years, due to the progress of neural networks, the deep learning and specifically deep reinforcement learning approaches [18,19] are becoming more and more popular.

The present article deals with the problem of stealth intrusion of UUV into the protected area in the conditions of a network-centric counteraction of the opposite side, equipped by the means of the detection of UUV. Following the work in Reference [6], the signal emitted by a mobile vehicle has a non-uniform space pattern. The TP problem of UUV’s evasion from detection by stationary sonar system (SSS) is considered. In the case of a passive mode of reception, the SSS detects UUV using a radiated signal generated by UUV’s motion [20]. An indicator of the success of the object’s stealth intrusion into the area is the integral risk of UUV detection on some assigned route [1,6]. When the TP problem is solved, the maximal UUV invisibility is achieved by choosing such trajectory and the law of velocity as time functions, which minimize the risk of detection. On the the other hand, an indicator of the effectiveness of the detection by SSS is the probability of detection of a moving UUV in the conditions of receiving a signal against random background noise of the aquatic environment [21]. Therefore equal values of detection risk assigned to given trajectory may match different values of detection probabilities depending on levels of a random background acoustic noise.

The current article expands and continues research presented in the article [6]. The need for new research emerged from a computer modeling of optimal UUV routes, conducted by the authors of the article using software, specially developed for this purpose. It turned out that for some types of radiation patterns the model trajectory has a sawtooth shape. This means that the UUV moves, constantly changing its heading angle. Accordingly, it is necessary to determine whether such trajectories are a feature of the developed numerical scheme for implementing the algorithm, or whether such trajectories are actually locally optimal in theory. A lot of new results have been revealed. It happens that the sawtooth trajectory shape appears due to violation of sufficient optimality conditions for extreme trajectories. In the current article new lemmas and theorems for the two-link and multi-link optimal trajectories are proven, as well as a lot of modeling examples are developed to illustrate all the obtained results. Additionally, special cases of radiation patterns leading to degeneration of necessary and sufficient conditions are studied. Moreover, the current research deals with the problem in a more general form, which allows studying of more various types of physical signals compared to the previous article [6].

The proposed work has the following structure. Section 2 discusses two statements of optimal TP problem as a two-pointed classical variation problem. The variables transformation, according to [22], is determined to reformulate the initial task to be more convenient for the further solution form. In Section 3 necessary optimality conditions are derived. Further in Section 4 sufficient optimality conditions are proved. Section 5 studies the conditions of the degeneration of Euler-Lagrange equations. The radiation patterns satisfying such conditions and the ones corresponding to zero Hessian are found. Section 6 proposes the extension of sufficient optimality conditions onto the class of two-link trajectories. Section 7 presents a number of examples, that illustrate and support analytical results obtained in the paper. Last Section 8 concludes the article and suggests a direction for future work.

## 2. Trajectory Planning Problem

The mobile vehicle moves on the plane in the detection region of the searching system, presented by one observer that is called a sensor. Let us formulate the problem of finding the optimal trajectories of the object as the variational problem with integral functional of object’s detection by the sensor, which is called risk of detection, or, for simplicity, just risk [6,10].

### 2.1. About the Risk Functional

The risk depends on the instantaneous level of physical signal S(t), radiated by the mobile vehicle and received by the sensor [6]. This signal is the function of the characteristics of the mobile vehicle and the sensor
(1)$$S={\left(\frac{v}{r}\right)}^{\mu }G\left(\varphi \right)\gamma \left(r,\varphi \right)g\left(\beta \right),$$
where multiplier G(φ) is responsible for the diagram of the sensor antenna, g(β) is the radiation pattern of the mobile vehicle in the direction toward the sensor which forms the angle π−β with velocity vector, *r* is the current distance between sensor and evading vehicle, *v*—absolute instant velocity of the object, γ(r,φ)—signal saturation factor in the physical medium of detection. The geometric meaning of angles ψ and φ is as follows. Here ψ is the angle of rotation of object’s velocity vector and φ is the angle of rotation of radius vector as shown in Figure 1a. Angle ψ−φ=β is the angle in triangle built from radial $v_{r}$ and transversal $v_{\varphi }$ velocities of the object, e.g., the angle between velocity vector of the vehicle and its projection on radius vector as shown in Figure 1b. The exponent μ characterizes the physical field used for detection. Depending on value μ this can be magnetic, thermal, acoustic or electromagnetic fields. Further in the paper the case of μ>1 will be considered.

For simplicity let us consider the sensor antenna diagram to be homogeneous, so G(φ)≡1, and that there is no additional signal attenuation or gain in the environment, so γ(r,φ)≡1.

The risk *R* is the integral value of the signal given by Equation (Equation 1), so the criterion of optimization is a function of phase coordinates of the vehicle and its relative disposition to the sensor:(2)$$R=\int_{0}^{T}{\left(\frac{v}{r}\right)}^{\mu }g\left(\beta \right)dt.$$

The example of radiation pattern is presented in Figure 2. The axes *X* and *Y* are oriented with the object in such a way that *Y* points to the upper side of the object, *X*-to the right one, and angle β is counted counter-clockwise from axes *X*. Function g(β) itself is a length of radius-vector for every β, thus g(β)>0.

### 2.2. Mathematical Statement of the Problem

The task of the mobile vehicle is to pass from start point *A* to final point *B* with as minimal risk on the trajectory as possible. The polar coordinate system (r,φ) in 2D space is introduced to rewrite the optimal trajectory planning problem.

As mentioned in the introduction, in [22] it was proven that for the problems of this type there is a transformation of coordinates which leads the functional to the convenient value for the next investigation form and to the coordinate space (ρ,φ), where ρ=lnr. It occurs that in this coordinate space optimal trajectories are geodetic, e.g., presented as straight lines. For the considered problem Equation (Equation 2) there is such a transformation too.

According to Lemma 1 [6] the original TP problem can be rewritten as the problem with the functional Equation (Equation 3).

**Problem** **1.**
*It is required to find the trajectory $\left(\rho^{*}\left(t\right),\varphi^{*}\left(t\right)\right)$, which minimizes the functional*
(3)$$R\left(\rho \left(\cdot \right),\varphi \left(\cdot \right)\right)=\int_{0}^{T}S\left(\rho ,\dot{\rho },\varphi ,\dot{\varphi },t\right)dt=\int_{0}^{T}{\left({\dot{\rho }}^{2}+{\dot{\varphi }}^{2}\right)}^{\mu /2}g\left(\arctan\frac{\dot{\varphi }}{\dot{\rho }}\right)dt⟶\min_{\rho (\cdot ),\varphi (\cdot )}.$$
*with boundary conditions*
$$\rho \left(0\right)=\rho_{A},\;\rho \left(T\right)=\rho_{B},\;\varphi \left(0\right)=\varphi_{A},\;\varphi \left(T\right)=\varphi_{B}.$$


Written in this form, the initial problem is more convenient for analytical solution, as the optimization functional depends only on the derivatives of coordinates.

## 3. The Necessary Optimality Conditions

First of all, the necessary conditions for extremal trajectory must be considered, e.g., the candidates for optimal trajectories are to be found. Due to the form of the Equation (Equation 3) this functional has the number of the first integrals. For example, as was shown in [6], Lagrangian $S(\rho ,\dot{\rho },\varphi ,\dot{\varphi },t)$ remain its value *S** along the extremal trajectory $\left(\rho^{*}\left(t\right),\varphi^{*}\left(t\right)\right)$.

Additionally, in [6] the theorem about the necessary condition of optimality for the trajectory was stated and proven. From Theorem 1 [6]

**Theorem** **1.**
*Suppose that $0<g_{1}<g\left(\beta \right)<g_{2}$ for all β∈[0,2π] is a twice continuously differentiated function of β, where $g_{1}$, $g_{2}$ are some constant values, and $\ddot{\rho }\left(t\right)$, $\ddot{\varphi }\left(t\right)$ exist and are continuous functions of t. Then the extremal trajectory satisfies the following system of equations*
(4)$$\left\{\begin{array}{c} \dot{\rho }=const,\\ \dot{\varphi }=const. \end{array}\right.$$


It follows that the extremal trajectory, which can be the solution for Problem 1 can be represented in the parametric form of logarithmic spiral
(5)$$\left\{\begin{array}{c} r\left(t\right)=r_{A}\exp\left(\frac{t}{T_{0}}\ln\frac{r_{B}}{r_{A}}\right),\\ \varphi \left(t\right)=\varphi_{A}+\frac{\varphi_{B}-\varphi_{A}}{T_{0}}t. \end{array}\right.$$

The explicit form of the UUV extremal trajectory equation on the plane is
(6)$$r\left(\varphi \right)=r_{A}\exp\left(\frac{\varphi -\varphi_{A}}{\varphi_{B}-\varphi_{A}}\ln\frac{r_{B}}{r_{A}}\right).$$

Next two lemmas define the velocity law of the vehicle on the extremal trajectory and the risk value on it.

**Lemma** **1.**
*The velocity law on the extremal trajectory Equation (Equation 6) is represented as follows*
(7)$$v\left(t\right)=\frac{r_{A}}{T}\exp\left(\frac{t}{T}\ln\frac{r_{B}}{r_{A}}\right)\sqrt{\ln^{2}\frac{r_{B}}{r_{A}}+{\left(\varphi_{B}-\varphi_{A}\right)}^{2}}.$$


We need to reformulate Lemma 4 from Reference [6] as follows

**Lemma** **2.**
*The explicit dependence risk (Equation 3) from boundary conditions on the extremal trajectory Equation (Equation 6) has the form*
(8)$$R^{*}=\frac{{\left({\left(\rho_{B}-\rho_{A}\right)}^{2}+{\left(\varphi_{B}-\varphi_{A}\right)}^{2}\right)}^{\mu /2}}{T^{\mu -1}}g\left(\beta^{*}\right)=\frac{L^{\mu }}{T^{\mu -1}}g\left(\beta^{*}\right),$$
*where $\beta^{*}=\arctan\frac{\varphi_{B}-\varphi_{A}}{\rho_{B}-\rho_{A}}$, and $L=\sqrt{{\left(\rho_{B}-\rho_{A}\right)}^{2}+{\left(\varphi_{B}-\varphi_{A}\right)}^{2}}$ is the length of the straight line segment between points A and B in the space (ρ,φ).*


The proof of Lemma 2 remains valid as shown in Reference [6].

## 4. The Sufficient Optimality Conditions

Now let us prove that obtained above extremal trajectory given by Equation (Equation 6) is the optimal solution to the Problem 1, e.g., it brings the strong minimum to the functional Equation (Equation 3). The Hessian matrix shall be investigated for that cause.

**Lemma** **3.**
*Let $S\left(\rho ,\dot{\rho },\varphi ,\dot{\varphi },t\right)={\left({\dot{\rho }}^{2}+{\dot{\varphi }}^{2}\right)}^{\mu /2}g\left(\beta \right)$, g(β) – thrice continuously differentiated function of β, then Hessian matrix H equals*
$$H=\;\left(\begin{array}{cc} H_{11} & H_{12}\\ H_{21} & H_{22} \end{array}\right)$$

*where*
(9)$$\begin{array}{c} H_{11}=\frac{\partial^{2}S}{\partial {\dot{\rho }}^{2}}=\left(-2\dot{\varphi }\dot{\rho }\left(\mu -1\right)g^{\prime }\left(\beta \right)+\mu \left({\dot{\varphi }}^{2}+{\dot{\rho }}^{2}\left(\mu -1\right)\right)g\left(\beta \right)+g^{\prime \prime }\left(\beta \right){\dot{\varphi }}^{2}\right){\left({\dot{\varphi }}^{2}+{\dot{\rho }}^{2}\right)}^{\frac{\mu }{2}-2},\\ H_{12}=\frac{\partial^{2}S}{\partial \dot{\rho }\partial \dot{\varphi }}=\left(\left({\dot{\rho }}^{2}-{\dot{\varphi }}^{2}\right)\left(\mu -1\right)g^{\prime }\left(\beta \right)+\dot{\rho }\dot{\varphi }\left(\left(\mu^{2}-2\mu \right)g\left(\beta \right)-g^{\prime \prime }\left(\beta \right)\right)\right){\left({\dot{\varphi }}^{2}+{\dot{\rho }}^{2}\right)}^{\frac{\mu }{2}-2},\\ H_{21}=\frac{\partial^{2}S}{\partial \dot{\varphi }\partial \dot{\rho }}=\left(\left({\dot{\rho }}^{2}-{\dot{\varphi }}^{2}\right)\left(\mu -1\right)g^{\prime }\left(\beta \right)+\dot{\rho }\dot{\varphi }\left(\left(\mu^{2}-2\mu \right)g\left(\beta \right)-g^{\prime \prime }\left(\beta \right)\right)\right){\left({\dot{\varphi }}^{2}+{\dot{\rho }}^{2}\right)}^{\frac{\mu }{2}-2},\\ H_{22}=\frac{\partial^{2}S}{\partial {\dot{\varphi }}^{2}}=\left(2\dot{\varphi }\dot{\rho }\left(\mu -1\right)g^{\prime }\left(\beta \right)+\mu \left({\dot{\rho }}^{2}+{\dot{\varphi }}^{2}\left(\mu -1\right)\right)g\left(\beta \right)+g^{\prime \prime }\left(\beta \right){\dot{\rho }}^{2}\right){\left({\dot{\varphi }}^{2}+{\dot{\rho }}^{2}\right)}^{\frac{\mu }{2}-2}, \end{array}$$
*and the Hessian itself is the determinant of the matrix*
(10)$$\det H={\left({\dot{\varphi }}^{2}+{\dot{\rho }}^{2}\right)}^{\mu -2}\left(\mu -1\right)\left(g^{2}\left(\beta \right)\mu^{2}+g\left(\beta \right)g^{\prime \prime }\left(\beta \right)\mu -g^{\prime 2}\left(\beta \right)\left(\mu -1\right)\right).$$


Next theorem gives the sufficient condition of extremal trajectory optimality.

**Theorem** **2.**
*Assume that the conditions of Theorem 1, Lemma 3 are satisfied, and the inequality detH>0 is valid for all values β. Then the optimal trajectory given by Equation (Equation 4) brings the strong minimum to the risk functional Equation (Equation 3).*


**Proof.** If the Hessian matrix is positively defined then the Legandre conditions are fulfilled. So second order minor is the same with Hessian itself and detH>0 by the theorem condition. Choose any first order minor $H_{11}$ and $H_{22}$ from Equation (Equation 9) and find its sign. At least one of them is positive because both values are fixed on the optimal trajectory and their sum equals
(11)$$\begin{array}{c} H_{11}+H_{22}={\left({\dot{\varphi }}^{2}+{\dot{\rho }}^{2}\right)}^{\frac{\mu }{2}-1}\left(g\left(\beta \right)\mu^{2}+g^{\prime \prime }\left(\beta \right)\right)=\\ =\frac{\frac{\det H}{\mu \left(\mu -1\right){\left({\dot{\varphi }}^{2}+{\dot{\rho }}^{2}\right)}^{\frac{\mu }{2}-1}}+{\left({\dot{\varphi }}^{2}+{\dot{\rho }}^{2}\right)}^{\frac{\mu }{2}-1}\left(g^{\prime 2}\left(\beta \right)\left(1-\frac{1}{\mu }\right)+\left(\mu^{2}-\mu \right)g^{2}\left(\beta \right)\right)}{g(\beta )}>0. \end{array}$$In a previous article [6] this theorem was considered for the particular case of μ=2. Of course, expression from Equation (Equation 11) equals the same from [6] after substituting this μ. Taking into account that μ>1 leads to the statement that both values $H_{11}$ and $H_{22}$ are strictly positive too.The Jacobin conditions have the form
(12)$$\begin{array}{c} \frac{d}{dt}\left(H_{11}\dot{\rho }+H_{12}\dot{\varphi }\right)=0,\\ \frac{d}{dt}\left(H_{21}\dot{\rho }+H_{22}\dot{\varphi }\right)=0, \end{array}$$
as $S_{\rho \dot{\rho }}=0$, $S_{\varphi \dot{\rho }}=0$, $S_{\rho \dot{\varphi }}=0$, $S_{\varphi \dot{\varphi }}=0$.Using minors of the Hessian matrix from Equation (Equation 9), the summands of equations from Equation (Equation 12) can be found
(13)$$H_{11}\dot{\rho }=\left(-2\dot{\varphi }{\dot{\rho }}^{2}\left(\mu -1\right)g^{\prime }\left(\beta \right)+\mu \dot{\rho }\left({\dot{\varphi }}^{2}+{\dot{\rho }}^{2}\left(\mu -1\right)\right)g\left(\beta \right)+g^{\prime \prime }\left(\beta \right)\dot{\rho }{\dot{\varphi }}^{2}\right){\left({\dot{\varphi }}^{2}+{\dot{\rho }}^{2}\right)}^{\frac{\mu }{2}-2},$$
(14)$$H_{12}\dot{\varphi }=\left(\dot{\varphi }\left({\dot{\rho }}^{2}-{\dot{\varphi }}^{2}\right)\left(\mu -1\right)g^{\prime }\left(\beta \right)+\dot{\rho }{\dot{\varphi }}^{2}\left(\left(\mu^{2}-2\mu \right)g\left(\beta \right)-g^{\prime \prime }\left(\beta \right)\right)\right){\left({\dot{\varphi }}^{2}+{\dot{\rho }}^{2}\right)}^{\frac{\mu }{2}-2},$$
(15)$$H_{21}\dot{\rho }=\left(\dot{\rho }\left({\dot{\rho }}^{2}-{\dot{\varphi }}^{2}\right)\left(\mu -1\right)g^{\prime }\left(\beta \right)+{\dot{\rho }}^{2}\dot{\varphi }\left(\left(\mu^{2}-2\mu \right)g\left(\beta \right)-g^{\prime \prime }\left(\beta \right)\right)\right){\left({\dot{\varphi }}^{2}+{\dot{\rho }}^{2}\right)}^{\frac{\mu }{2}-2},$$
(16)$$H_{22}\dot{\varphi }=\left(2{\dot{\varphi }}^{2}\dot{\rho }\left(\mu -1\right)g^{\prime }\left(\beta \right)+\mu \dot{\varphi }\left({\dot{\rho }}^{2}+{\dot{\varphi }}^{2}\left(\mu -1\right)\right)g\left(\beta \right)+g^{\prime \prime }\left(\beta \right){\dot{\rho }}^{2}\dot{\varphi }\right){\left({\dot{\varphi }}^{2}+{\dot{\rho }}^{2}\right)}^{\frac{\mu }{2}-2}.$$Substituting Equations (Equation 13)–(Equation 16) to Equation (Equation 12), one can get
$$\begin{array}{c} H_{12}\dot{\varphi }+H_{11}\dot{\rho }=\left(\left(\mu -1\right)g^{\prime }\left(\beta \right)\left({\dot{\rho }}^{2}\dot{\varphi }-{\dot{\varphi }}^{3}-2\dot{\varphi }{\dot{\rho }}^{2}\right)+g\left(\beta \right)\mu \dot{\rho }\left({\dot{\varphi }}^{2}\left(\mu -2\right)+{\dot{\varphi }}^{2}+{\dot{\rho }}^{2}\left(\mu -1\right)\right)\right.+\\ +\left.g^{\prime \prime }\left(\beta \right)\left(-\dot{\rho }{\dot{\varphi }}^{2}+\dot{\rho }{\dot{\varphi }}^{2}\right)\right){\left({\dot{\varphi }}^{2}+{\dot{\rho }}^{2}\right)}^{\frac{\mu }{2}-2}=\left(-\left(\mu -1\right)g^{\prime }\left(\beta \right)\dot{\varphi }\left({\dot{\rho }}^{2}+{\dot{\varphi }}^{2}\right)+g\left(\beta \right)\mu \dot{\rho }\left(\mu -1\right)\left({\dot{\rho }}^{2}+{\dot{\varphi }}^{2}\right)\right)*\\ {}*{\left({\dot{\varphi }}^{2}+{\dot{\rho }}^{2}\right)}^{\frac{\mu }{2}-2}=\left(-g^{\prime }\left(\beta \right)\dot{\varphi }+g\left(\beta \right)\mu \dot{\rho }\right)\left(\mu -1\right){\left({\dot{\varphi }}^{2}+{\dot{\rho }}^{2}\right)}^{\frac{\mu }{2}-1}=\left(\mu -1\right)S_{\dot{\rho }}, \end{array}$$
$$\begin{array}{c} H_{22}\dot{\varphi }+H_{21}\dot{\rho }=\left(\dot{\rho }\left(\mu -1\right)g^{\prime }\left(\beta \right)\left({\dot{\rho }}^{2}+{\dot{\varphi }}^{2}\right)+\mu g\left(\beta \right)\dot{\varphi }\left({\dot{\rho }}^{2}+{\dot{\varphi }}^{2}\left(\mu -1\right)+{\dot{\rho }}^{2}\left(\mu -2\right)\right)\right){\left({\dot{\varphi }}^{2}+{\dot{\rho }}^{2}\right)}^{\frac{\mu }{2}-2}=\\ =\left(\dot{\rho }g^{\prime }\left(\beta \right)+\mu g\left(\beta \right)\dot{\varphi }\right)\left(\mu -1\right){\left({\dot{\varphi }}^{2}+{\dot{\rho }}^{2}\right)}^{\frac{\mu }{2}-1}=\left(\mu -1\right)S_{\dot{\varphi }}. \end{array}$$The Equation (Equation 12) takes the form
$$\begin{array}{c} \frac{d}{dt}S_{\dot{\rho }}=0,\\ \frac{d}{dt}S_{\dot{\varphi }}=0, \end{array}$$
which coincides with explicit form of Euler–Lagrange Equations (Equation 17) for considered Problem 1.
(17)$$\left\{\begin{array}{c} \mu \frac{\dot{\rho }}{{\dot{\rho }}^{2}+{\dot{\varphi }}^{2}}S^{*}-\frac{\dot{\varphi }}{{\dot{\rho }}^{2}+{\dot{\varphi }}^{2}}{\left(\ln g(\beta )\right)}^{\prime }S^{*}=C_{1},\\ \mu \frac{\dot{\varphi }}{{\dot{\rho }}^{2}+{\dot{\varphi }}^{2}}S^{*}+\frac{\dot{\rho }}{{\dot{\rho }}^{2}+{\dot{\varphi }}^{2}}{\left(\ln g(\beta )\right)}^{\prime }S^{*}=C_{2}. \end{array}\right.$$Here $C_{1}$, $C_{2}$, *S** are constant values, representing three first integrals of Euler–Lagrange system. It means that the sufficient conditions of strong minimum of risk functional Equation (Equation 3) are fulfilled [23]. □

Thus, positivity of the determinant of the Hessian matrix guarantees the logarithmic spiral to be optimal solution of the problem. Theorem 2 allows for any g(β) to answer if extremal trajectory (Equation 6) is optimal or not. Indeed, the sign of Hessian (Equation 10) depends on the sign of the third multiplier, which depends only on the function g(β). If in conditions of Theorem 2 detH≤0, then sufficient optimality conditions are not fulfilled and the optimal trajectory may have a more complicated form and consist of several logarithmic spirals. These cases are discussed in the paper further.

## 5. Degeneration of Euler–Lagrange Equations and Zero Hessian Cases

First of all the case of degeneration of Euler–Lagrange Equations (Equation 17) should be considered. The thing is that both equations may coincide to each other for some radiation pattern g(β) and some constant values $C_{1}$, $C_{2}$.

**Lemma** **4.**
*The function $g\left(\beta \right)={C|\cos\left(\gamma -\beta \right)|}^{\mu }$, where $\tan\gamma =\frac{C_{2}}{C_{1}}$, brings the Euler–Lagrange system (Equation 17) to degeneration.*


**Proof.** Let us multiply the first equation from system (Equation 17) by $\dot{\rho }$, the second one by $\dot{\varphi }$ and then sum these two new equations up. Similarly, let us multiply the first equation from system (Equation 17) by $-\dot{\varphi }$, the second one by $\dot{\rho }$ and then sum these two new equations up. A new system of equations appears
(18)$$\left\{\begin{array}{c} C_{1}\dot{\rho }+C_{2}\dot{\varphi }=\mu S^{*},\\ C_{2}\dot{\rho }-C_{1}\dot{\varphi }=S^{*}{\left(\ln g(\beta )\right)}^{\prime }. \end{array}\right.$$Dividing the second equation from system of Equations (Equation 18) on the first one can obtain
(19)$${\left(\ln g(\beta )\right)}^{\prime }=\mu \frac{C_{2}\dot{\rho }-C_{1}\dot{\varphi }}{C_{1}\dot{\rho }+C_{2}\dot{\varphi }}=\mu \tan\left(\gamma -\beta \right),$$
where $\tan\gamma =\frac{C_{2}}{C_{1}}$ and is some constant value, $\tan\beta =\frac{\dot{\varphi }}{\dot{\rho }}$ and depends from vehicle velocity direction. Integration of the right part of Equation (Equation 19) on β gives
(20)$$\int \mu \tan\left(\gamma -\beta \right)d\beta =\mu \int \frac{\sin(\gamma -\beta )}{\cos(\gamma -\beta )}d\beta =\mu \int \frac{d(\cos(\gamma -\beta ))}{\cos(\gamma -\beta )}d\beta =\mu \ln\left|\cos\left(\gamma -\beta \right)\right|+\tilde{C}$$
with undefined constant of integration $\tilde{C}$. Now integrating left part of Equation (Equation 19) one can obtain
(21)$$\ln g\left(\beta \right)=\mu \ln|\cos\left(\gamma -\beta \right)|+\tilde{C}.$$Using both parts of Equation (Equation 21) as powers for the exponent leads to the statement of the Lemma, where $C=\exp\tilde{C}$ and is some constant value. □

According to Lemma 4 obtained special form of function g(β) degenerates Euler–Lagrange equations only with one specific value of the parameter γ, that responds to the direction associated with boundary conditions of the problem 1 expressed through constants $C_{1}$ and $C_{2}$ when it is already solved.

**Corollary** **1.**
*The function g(β) from Lemma 4 brings the Lagrangian of the functional (Equation 3) to the form*
(22)$$S\left(\rho ,\dot{\rho },\varphi ,\dot{\varphi },t\right)={\left({\tilde{C}}_{1}\dot{\rho }+{\tilde{C}}_{2}\dot{\varphi }\right)}^{\mu }.$$


Proof of Corollary 1 is given in Appendix A.

In addition to Theorem 1 the next theorem presents the form of extremal trajectories now for the special type of radiation pattern from Lemma 4.

**Theorem** **3.**
*All the curves satisfying equation ${\tilde{C}}_{1}\dot{\rho }+{\tilde{C}}_{2}\dot{\varphi }=C_{3}$ on the coordinate plane (ρ,φ) with some constants ${\tilde{C}}_{1}$, ${\tilde{C}}_{2}$, $C_{3}$ are extremal when g(β) satisfies Lemma 4.*


**Proof.** It is enough to remember Corollary 1 to state that *S* is a constant on the extremal trajectory locally in vicinity directions β where g(β) is continuously differentiated function. Then from Equation (Equation 22) it follows the theorem’s statement. □

So, among the set of optimal paths leading from *A* to *B* that satisfy the Equation (Equation 22), there is also the logarithmic spiral shown in Equation (Equation 6) with the same value of the risk function.

After analysis of degeneration of necessary optimality conditions it is time to consider the sufficient conditions too. One of the interesting for research cases is the case of zero Hessian (Equation 10)
(23)$$g^{2}\left(\beta \right)\mu^{2}+g\left(\beta \right)g^{\prime \prime }\left(\beta \right)\mu -g^{\prime 2}\left(\beta \right)\left(\mu -1\right)=0.$$

**Lemma** **5.**
*The function g(β) which nulls the Hessian (Equation 10) equals*
(24)$$g\left(\beta \right)=C|\cos\left(\tilde{\beta }-\beta \right){|}^{\mu },$$
*where $\tilde{\beta }$ is an arbitrary constant.*


**Proof.** Dividing of Equation (Equation 23) by $g^{\prime }\left(\beta \right)g\left(\beta \right)$ brings it to the form:
(25)$$\frac{g^{\prime \prime }\left(\beta \right)}{g^{\prime }\left(\beta \right)}\mu -\frac{g^{\prime }\left(\beta \right)}{g(\beta )}\left(\mu -1\right)+\mu^{2}\frac{g(\beta )}{g^{\prime }\left(\beta \right)}=0.$$Introducing a new variable $\eta =\frac{g^{\prime }\left(\beta \right)}{g(\beta )}$ allows us to rewrite Equation (Equation 25) in a form
(26)$$\eta^{\prime }\mu +\eta^{2}+\mu^{2}=0.$$Solving Equation (Equation 26) gives
$$\eta =\mu \tan(-\beta +\tilde{\beta }),$$
where $\tilde{\beta }$ is an arbitrary constant. After returning to function g(β) this equation becomes
$$\frac{g^{\prime }\left(\beta \right)}{g(\beta )}=\mu \tan\left(-\beta +\tilde{\beta }\right)$$Addressing the proof of Lemma 4 and Equation (Equation 19) it is fair to state that
$$g\left(\beta \right)=C|\cos\left(\tilde{\beta }-\beta \right){|}^{\mu }.$$□

Now comparing radiation pattern g(β) given by Lemma 4 and the one settled by Lemma 5 we have the identity of both expressions at fixed γ and some value of $\beta_{0}$. Numerical modelling for the zero Hessian case will be held in Section 7.

## 6. Two-Link Trajectories

This section considers optimal trajectories for radiation patterns which do not satisfy sufficient conditions from Section 4. For simplicity and analytical solvability of the problems only two-link trajectories are considered. The risk on the optimal two-link trajectory passing through points *A*, *B*, *C* is given by the next Lemma.

**Lemma** **6.**
*The minimum value of risk on a two-segment trajectory that passes through the points A, C, B in the time interval T is equal to*
(27)$$R_{ACB}=\frac{1}{T^{\mu -1}}{\left(L_{1}g^{1/\mu }\left(\beta_{1}\right)+L_{2}g^{1/\mu }\left(\beta_{2}\right)\right)}^{\mu },$$
*where*
(28)$$\begin{array}{c} L_{1}=\sqrt{{\left(\rho_{C}-\rho_{A}\right)}^{2}+{\left(\varphi_{C}-\varphi_{A}\right)}^{2}},\\ L_{2}=\sqrt{{\left(\rho_{B}-\rho_{C}\right)}^{2}+{\left(\varphi_{B}-\varphi_{C}\right)}^{2}},\\ \beta_{1}=\arctan\left(\frac{\varphi_{C}-\varphi_{A}}{\rho_{C}-\rho_{A}}\right),\\ \beta_{2}=\arctan\left(\frac{\varphi_{B}-\varphi_{C}}{\rho_{B}-\rho_{C}}\right). \end{array}$$


The proof of Lemma 6 is proposed in Appendix A.

Lemma 6 allows us to check the type of optimal trajectory. If the risk on the two-link trajectory is less than on the logarithmic spiral directly connecting the points *A*, *B*, then two-link trajectory is optimal.

As was stated in [22], in (ρ,φ) space logarithmic spirals are presented as straight lines. Considering optimal two-link trajectories in (ρ,φ) space (Figure 3) allows to rewrite Equation (Equation 27) in a simpler way.

**Corollary** **2.**
*The minimum value of risk on a two-segment trajectory that passes through the points A, C, B in the time interval T can be represented as*
(29)$$R_{ACB}=\frac{L_{0}^{\mu }}{T^{\mu -1}}{\left(\frac{\sin\left(\beta_{2}-\beta_{0}\right)g^{1/\mu }\left(\beta_{1}\right)+\sin\left(\beta_{0}-\beta_{1}\right)g^{1/\mu }\left(\beta_{2}\right)}{\sin(\beta_{2}-\beta_{1})}\right)}^{\mu },$$
*where*
(30)$$\begin{array}{c} L_{0}=\sqrt{{\left(\rho_{B}-\rho_{A}\right)}^{2}+{\left(\varphi_{B}-\varphi_{A}\right)}^{2}},\\ \beta_{1}=\arctan\left(\frac{\varphi_{C}-\varphi_{A}}{\rho_{C}-\rho_{A}}\right),\\ \beta_{2}=\arctan\left(\frac{\varphi_{B}-\varphi_{C}}{\rho_{B}-\rho_{C}}\right),\\ \beta_{0}=\arctan\left(\frac{\varphi_{B}-\varphi_{A}}{\rho_{B}-\rho_{A}}\right). \end{array}$$


The proof of Corollary 2 is given in Appendix A.

A few words should be said about the applicability of the formula (Equation 29). The expression under parentheses must always be positive. Indeed, as shown in Figure 3, either an inequality $-\pi /2+\beta_{0}\leq \beta_{1}\leq \beta_{0}\leq \beta_{2}\leq \pi /2+\beta_{0}$ or an inequality $-\pi /2+\beta_{0}\leq \beta_{2}\leq \beta_{0}\leq \beta_{1}\leq \pi /2+\beta_{0}$ is satisfied due to the geometry. Because g(β)>0 in both cases this expression is positive.

According to proof of Lemma 6 the value of risk on two-link extremal trajectory equals
(31)$$R_{ACB}\left(\beta_{1},\beta_{2},T_{1},T_{2}\right)=\frac{L_{1}^{\mu }\left(\beta_{1}\right)}{T_{1}^{\mu -1}}g\left(\beta_{1}\right)+\frac{L_{1}^{\mu }\left(\beta_{2}\right)}{T_{2}^{\mu -1}}g\left(\beta_{2}\right).$$

**Problem** **2.**
*It is needed to find a quad $\left(\beta_{1}^{*},\beta_{2}^{*},T_{1}^{*},T_{2}^{*}\right)$ that*
(32)$$\left(\beta_{1}^{*},\beta_{2}^{*},T_{1}^{*},T_{2}^{*}\right)=\arg\min_{\beta_{1},\beta_{2},\;T_{1},T_{2},\;T_{1}+T_{2}=T\geq 0}R_{ACB}\left(\beta_{1},\beta_{2},T_{1},T_{2}\right).$$


Instead of solving the optimization Problem 2 of finding the minimum of $R_{ACB}$$\left(\beta_{1},\beta_{2},T_{1},T_{2}\right)$ for four variables, we will solve two consecutive optimization Problem 3 for pairs ($T_{1}$, $T_{2}$) and ($\beta_{1}$, $\beta_{2}$). This is possible because Lemma 6 found the unique pair $T_{1}$ and $T_{2}$ that gives the minimum.

**Problem** **3.**
*It is needed to find a quad $\left(\beta_{1}^{*},\beta_{2}^{*},T_{1}^{*},T_{2}^{*}\right)$ that gives consecutive minimum of $R_{ACB}\left(\beta_{1},\beta_{2},T_{1},T_{2}\right)$, so*
(33)$$R_{ACB}\left(\beta_{1},\beta_{2},T_{1},T_{2}\right)\to \min_{\beta_{1},\beta_{2}}\left(\min_{T_{1},T_{2}\geq 0,\;T_{1}+T_{2}=T}\right).$$


Next theorem states a result about the structure of the optimal two-link trajectories.

**Theorem** **4.**
*Optimal angles $\beta_{1}^{*},\beta_{2}$* for two-link optimal trajectory do not depend on the direction $\beta_{0}$ between start and end points, are fully defined by radiation pattern g(β) and satisfy*
(34)$$\left\{\begin{array}{c} {\left(\frac{g(\beta_{2})}{g(\beta_{1})}\right)}^{1/\mu }=\cos\left(\beta_{2}-\beta_{1}\right)+\frac{1}{\mu }\sin\left(\beta_{2}-\beta_{1}\right)\frac{g^{\prime }\left(\beta_{1}\right)}{g(\beta_{1})},\\ {\left(\frac{g(\beta_{1})}{g(\beta_{2})}\right)}^{1/\mu }=\cos\left(\beta_{2}-\beta_{1}\right)-\frac{1}{\mu }\sin\left(\beta_{2}-\beta_{1}\right)\frac{g^{\prime }\left(\beta_{2}\right)}{g(\beta_{2})}. \end{array}\right.$$


**Proof.** To prove this theorem an expression from (Equation 29) must be investigated. After omitting non-dependable on $\beta_{0}$ constants $L_{0}$ and *T* the variable $\tilde{R}=R^{\mu }$ can be considered
(35)$$\tilde{R}=\frac{\sin\left(\beta_{2}-\beta_{0}\right)g^{1/\mu }\left(\beta_{1}\right)+\sin\left(\beta_{0}-\beta_{1}\right)g^{1/\mu }\left(\beta_{2}\right)}{\sin(\beta_{2}-\beta_{1})}.$$This formula should be minimized by $\beta_{1}$ and $\beta_{2}$, e.g.,
(36)$$\left\{\begin{array}{c} \frac{\partial R}{\partial \beta_{1}}=0,\\ \frac{\partial R}{\partial \beta_{2}}=0. \end{array}\right.$$The first equation of System (Equation 36) leads to the expression
$$\begin{array}{c} \frac{\partial R}{\partial \beta_{1}}=\frac{\left(\sin\left(\beta_{2}-\beta_{0}\right)\frac{1}{\mu }g^{\prime }\left(\beta_{1}\right)g^{1/\mu -1}\left(\beta_{1}\right)-\cos\left(\beta_{0}-\beta_{1}\right)g^{1/\mu }\left(\beta_{2}\right)\right)\sin\left(\beta_{2}-\beta_{1}\right)}{\sin^{2}\left(\beta_{2}-\beta_{1}\right)}\\ +\frac{\cos\left(\beta_{2}-\beta_{1}\right)\left(\sin\left(\beta_{2}-\beta_{0}\right)g^{1/\mu }\left(\beta_{1}\right)+\sin\left(\beta_{0}-\beta_{1}\right)g^{1/\mu }\left(\beta_{2}\right)\right)}{\sin^{2}\left(\beta_{2}-\beta_{1}\right)}=0. \end{array}$$After simplifying the last equation becomes
$$\begin{array}{c} \frac{\partial R}{\partial \beta_{1}}=g^{1/\mu }\left(\beta_{1}\right)\sin\left(\beta_{2}-\beta_{0}\right)\frac{\frac{1}{\mu }\sin\left(\beta_{2}-\beta_{1}\right)\frac{g^{\prime }\left(\beta_{1}\right)}{g(\beta_{1})}+\cos\left(\beta_{2}-\beta_{1}\right)}{\sin^{2}\left(\beta_{2}-\beta_{1}\right)}-g^{1/\mu }\left(\beta_{2}\right)\frac{\sin(\beta_{2}-\beta_{0})}{\sin^{2}\left(\beta_{2}-\beta_{1}\right)}=0. \end{array}$$This equation may be presented as a product of two multipliers so one equation from the next list of two equations is fulfilled
(37)$$\left.\begin{array}{c} \sin(\beta_{2}-\beta_{0})=0,\\ g^{1/\mu }\left(\beta_{2}\right)=g^{1/\mu }\left(\beta_{1}\right)\left(\cos\left(\beta_{2}-\beta_{1}\right)+\frac{1}{\mu }\sin\left(\beta_{2}-\beta_{1}\right)\frac{g^{\prime }\left(\beta_{1}\right)}{g(\beta_{1})}\right). \end{array}\right.$$The first equation corresponds to the case of one-link extremal trajectory, so it does not represent an interest.Similarly the second equation of System (Equation 36) leads to the expression
$$\begin{array}{c} \frac{\partial R}{\partial \beta_{2}}=\frac{\left(\cos\left(\beta_{2}-\beta_{0}\right)g^{1/\mu }\left(\beta_{1}\right)+\sin\left(\beta_{0}-\beta_{1}\right)\frac{1}{\mu }g^{\prime }\left(\beta_{2}\right)g^{1/\mu -1}\left(\beta_{2}\right)\right)\sin\left(\beta_{2}-\beta_{1}\right)}{\sin^{2}\left(\beta_{2}-\beta_{1}\right)}\\ -\frac{\cos\left(\beta_{2}-\beta_{1}\right)\left(\sin\left(\beta_{2}-\beta_{0}\right)g^{1/\mu }\left(\beta_{1}\right)+\sin\left(\beta_{0}-\beta_{1}\right)g^{1/\mu }\left(\beta_{2}\right)\right)}{\sin^{2}\left(\beta_{2}-\beta_{1}\right)} \end{array}$$After simplifying it becomes
$$\begin{array}{c} \frac{\partial R}{\partial \beta_{2}}=g^{1/\mu }\left(\beta_{1}\right)\frac{\sin(\beta_{0}-\beta_{1})}{\sin^{2}\left(\beta_{2}-\beta_{1}\right)}+g^{1/\mu }\left(\beta_{2}\right)\sin\left(\beta_{0}-\beta_{1}\right)\frac{\sin\left(\beta_{2}-\beta_{1}\right)\frac{1}{\mu }\frac{g^{\prime }\left(\beta_{2}\right)}{g(\beta_{2})}-\cos\left(\beta_{2}-\beta_{1}\right)}{\sin^{2}\left(\beta_{2}-\beta_{1}\right)}=0. \end{array}$$Again, a list of two equations is fair where at least one of them is fulfilled
(38)$$\left.\begin{array}{c} \sin(\beta_{0}-\beta_{1})=0,\\ g^{1/\mu }\left(\beta_{1}\right)=g^{1/\mu }\left(\beta_{2}\right)\left(\cos\left(\beta_{2}-\beta_{1}\right)-\frac{1}{\mu }\sin\left(\beta_{2}-\beta_{1}\right)\frac{g^{\prime }\left(\beta_{2}\right)}{g(\beta_{2})}\right). \end{array}\right.$$The first equation corresponds to the case of one-segment trajectory, because it is already explored.The second equations of Lists (Equation 37) and (Equation 38) form a new System (Equation 34). The explicit dependence $\beta_{1},\beta_{2}$ from $\beta_{0}$ is absent. □

So, any optimal two-link trajectory for specific radiation pattern g(β) is constructed from segments of two optimal directions $\beta_{1}$* and $\beta_{2}$*. Moreover, it can be shown, that optimal multi-link trajectories are based on these segments as well.

The next lemma specifies a system of equations derived in Theorem 4, which allows us to find optimal angles $\beta_{1}$* and $\beta_{2}$* that minimize a risk functional on two-link trajectory.

**Lemma** **7.**
*Angles $\beta_{1}^{*},\beta_{2}$* for optimal trajectory can be found from system*
(39)$$\left\{\begin{array}{c} \cos\left(\beta_{2}-\beta_{1}\right)=\cos\left(\xi_{1}-\xi_{2}\right),\\ \frac{g(\beta_{2})}{g(\beta_{1})}=\left|\frac{\cos^{\mu }\xi_{2}}{\cos^{\mu }\xi_{1}}\right|, \end{array}\right.$$
*where*
(40)$$\left\{\begin{array}{c} \xi_{1}=\arctan\frac{1}{\mu }\frac{g^{\prime }\left(\beta_{1}\right)}{g(\beta_{1})},\\ \xi_{2}=\arctan\frac{1}{\mu }\frac{g^{\prime }\left(\beta_{2}\right)}{g(\beta_{2})}. \end{array}\right.$$


The proof of Lemma 7 is proposed in Appendix A.

Let us ask the question of what value the risk will take in the case when the radiation pattern corresponds to the case of zero Hessian.

**Lemma** **8.**
*Radiation pattern $g\left(\beta \right)=C|\cos\left(\tilde{\beta }-\beta \right){|}^{\mu }$ makes two-link trajectory risk a constant value for any $\beta_{1},\;\beta_{2}\in \left[-\pi /2+\tilde{\beta },\pi /2+\tilde{\beta }\right]$.*


**Proof.** Substituting this radiation pattern into Equation (Equation 29) and getting rid of the module in function g(β) gives
$$R=\frac{L_{0}^{\mu }C^{\mu }}{T^{\mu -1}}{\left(\frac{\sin\left(\beta_{2}-\beta_{0}\right)\cos\left(\tilde{\beta }-\beta_{1}\right)+\sin\left(\beta_{0}-\beta_{1}\right)\cos\left(\tilde{\beta }-\beta_{2}\right)}{\sin(\beta_{2}-\beta_{1})}\right)}^{\mu },$$Simplifying this expression leads to the chain of equations
$$R=\frac{L_{0}^{\mu }C^{\mu }}{T^{\mu -1}}{\left(\frac{\sin\left(\beta_{2}-\beta_{0}-\tilde{\beta }+\beta_{1}\right)+\sin\left(\beta_{2}-\beta_{0}+\tilde{\beta }-\beta_{1}\right)+\sin\left(\beta_{0}-\beta_{1}-\tilde{\beta }+\beta_{2}\right)+\sin\left(\beta_{0}-\beta_{1}+\tilde{\beta }-\beta_{2}\right)}{2\sin(\beta_{2}-\beta_{1})}\right)}^{\mu },$$
$$R=\frac{L_{0}^{\mu }C^{\mu }}{T^{\mu -1}}{\left(\frac{\sin\left(\beta_{2}-\beta_{0}+\tilde{\beta }-\beta_{1}\right)+\sin\left(\beta_{0}-\beta_{1}-\tilde{\beta }+\beta_{2}\right)}{2\sin(\beta_{2}-\beta_{1})}\right)}^{\mu },$$
$$R=\frac{L_{0}^{\mu }C^{\mu }}{T^{\mu -1}}{\left(\frac{\sin\left(\beta_{2}-\beta_{1}\right)\cos\left(\tilde{\beta }-\beta_{0}\right)}{\sin(\beta_{2}-\beta_{1})}\right)}^{\mu },$$
(41)$$R=\frac{L_{0}^{\mu }C^{\mu }}{T^{\mu -1}}\cos^{\mu }\left(\tilde{\beta }-\beta_{0}\right)={\left(\frac{L_{0}C}{T}\cos\left(\tilde{\beta }-\beta_{0}\right)\right)}^{\mu }T.$$From here it follows that *R* is a constant value for any $\beta_{1},\;\beta_{2}\in \left[-\pi /2+\tilde{\beta },\pi /2+\tilde{\beta }\right]$ and depends only on boundary conditions $L_{0},\;\beta_{0}$ and fixed time on route *T*. Moreover, the function of radiation pattern from Lemma 4 makes expression from Equation (Equation 41) even simpler, as in this case $\tilde{\beta }=\beta_{0}$
$$R={\left(\frac{L_{0}C}{T}\right)}^{\mu }T.$$□

## 7. Examples and Illustration of the Results

The current article develops the analytical methods for the constructing of optimal reference trajectories. However, in the first part of work [6] an explicit form criterion for the PP problem for various noise conditions and two types of decision rules are given, based on a comparison of threshold statistics. The real-world data describing these statistics can be used for practical applications of research introduced in these articles. Detection probabilities can be obtained through the calculation of the risk criterion. Certainly, practical aspects make all the analytical models and estimations much more complicated, but still the presented in the article research can be helpful for finding reference paths for UUVs.

Matlab scripts have been developed to simulate, validate numerically found analytical solutions and support lemmas and theorems. Each of the examples below demonstrates results for different chosen radiation patterns and illustrates one of the cases discussed throughout the article. In each example the start point *A* has coordinates (50, 0) and end point B—(0,100) on Cartesian plane, so $\rho_{0}=\ln50$, $\varphi_{0}=0$. Sensor *S* is situated in the origin.

### 7.1. Example 1. Logarithmic Spiral

The first example considers a simple case of positive Hessian on the whole range of $\beta \in [0^{\circ },360^{\circ }]$ or β∈[0,2π] (here and further in some examples angle β may be also considered in degrees), or on the range of $\beta \in [0^{\circ },180^{\circ }]$ or β∈[0,π] due to symmetry g(β), with radiation pattern
(42)$$g\left(\beta \right)=K_{1}+K_{2}\cos^{2}\left(\beta \right),$$
where $K_{1}=0.25$, $K_{2}=0.75$. Radiation pattern described by Equation (Equation 42) was considered in [6], but in the case of μ=2, when detH=1 and is constant due to Lemma 3. However, now this is not the same case, as μ=3.

Figure 4a shows a radiation pattern as a function of angle g(β) and Figure 4b—as a curve on Cartesian plane in respect to the mobile vehicle, where g(β) is a length of the radius-vector. Figure 5 shows the dependence of the Hessian detH(β). As one can see, it is positive for all angles, thus the sufficient condition of Theorem 2 is fulfilled, meaning that logarithmic spiral AB is the optimal solution with minimal risk on trajectory risk on the trajectory R=0.5487.

This fact is illustrated in Figure 6a and Figure 6b. Figure 6a shows the surface of the risk above $\left(\rho -\rho_{0},\;\varphi -\varphi_{0}\right)$ plane, e.g., the vertical axis is the value of risk on two-link trajectory with fixed start and end points *A* and *B* and switch point *C* with coordinates $\left(\rho -\rho_{0},\;\varphi -\varphi_{0}\right)$. Figure 6b illustrates the projection of this surface on the plane $\left(\rho -\rho_{0},\;\varphi -\varphi_{0}\right)$. The black color shows the optimal trajectory AB, which is a line segment, as mentioned in [22].

Figure 7 demonstrates the found optimal trajectory on Cartesian plane.

### 7.2. Example 2. Two-Link Optimal Trajectory

The second example illustrates a case of two-link optimal trajectory. As in previous example, μ=3. Radiation pattern is more complicated and, as shown in Figure 8, is presented as an ellipse with cosine wave
$$g\left(\beta \right)=\frac{1}{\sqrt{1+0.8\sin(\beta )}}+0.3\cos\left(6\beta \right).$$

Figure 9 shows that, in contrast to Example 1, Hessian detH is not positive for all β’s, that means that sufficient conditions are not satisfied. Both one-link AB and two-link trajectories $AC_{1}B$, $AC_{2}B$ are shown in Figure 10 and Figure 11. A programming module in Matlab has been developed to compute optimal $\beta_{1}$*, $\beta_{2}$* based on system of Equation (Equation 39) from Lemma 7. Route points coordinates are found $C_{1}=\left(\rho_{C_{1}},\varphi_{C_{1}}\right)=\left(4.012,1.139\right)$, $C_{2}=\left(\rho_{C_{2}},\varphi_{C_{2}}\right)=\left(4.508,0.428\right)$. Trajectories $AC_{1}B$ and $AC_{2}B$ are symmetrical in a sense of angles $\beta_{1}$* and $\beta_{2}$*: $\beta_{1}^{*}=84.9836^{\circ }$ corresponds to segments $AC_{1}$ and $C_{2}B$, $\beta_{2}^{*}=35.6791^{\circ }$–to segments $C_{1}B$ and $AC_{2}$. The values of found risks on these trajectories are
$$\begin{array}{c} R_{AB}=1.3459,\\ R_{AC_{1}B}=R_{AC_{2}B}=0.9762. \end{array}$$

Obviously, the two-link paths are preferable in a sense of risk minimization by almost 37%. In this example a unique pair $\left(\beta_{1}^{*},\beta_{2}^{*}\right)$ is a solution of Problem 3.

### 7.3. Example 3. The Case of Null Hessian

The third example is devoted to the case of radiation pattern, described in Lemma 5 of Section 5, e.g., the function g(β) that nulls the Hessian. We choose in this example μ=2. The radiation pattern is presented on Figure 12 and has a form
$$g\left(\beta \right)=25\cos^{2}\left(\gamma +\beta \right),\;\;\beta \in \left[0,\pi /2\right],$$
where tanγ=0.75, and g(β)=g(−β)=g(π−β).

Figure 13 confirms the zeroing of Hessian for given radiation pattern at each of the intervals $\beta \in (0^{\circ },90^{\circ })$, $\beta \in (90^{\circ },180^{\circ })$, $\beta \in (180^{\circ },270^{\circ })$, $\beta \in (270^{\circ },360^{\circ })$.

Figure 14 demonstrates the validity of Lemma 8, as it is clear from the risk surface, that risk values of all two-link trajectories lie on a plateau, e.g., they are constant. That means that any two-link trajectory has the same risk value, which equals R=34.9668.

### 7.4. Example 4. The Case of Some Extremes of Risk Function

The fourth example considers a complex radiation pattern, shown in Figure 15. In this example μ=2.

The peculiarity of this case and chosen radiation pattern can be observed in Figure 16, Figure 17 and Figure 18 —the risk surface contains not only local minimums, but also local maximums, that can be obtained from Theorem 4 and Lemma 8. Trajectories $AC_{3}B$ and $AC_{4}B$ are two-link trajectories which pass through points $C_{3}$ and $C_{4}$ of the risk function local maximum on variables $\left(\beta_{1},\beta_{2}\right)$ for Problem 3.

Route points coordinates are found: $C_{1}=\left(\rho_{C_{1}},\varphi_{C_{1}}\right)=\left(3.9259,1.1\right)$, $C_{2}=\left(\rho_{C_{2}},\varphi_{C_{2}}\right)=\left(4.5947,0.4712\right)$. Optimal angles $\beta_{1}$* and $\beta_{2}$*: $\beta_{1}^{*}=89.28^{\circ }$, $\beta_{2}^{*}=34.6145^{\circ }$. The values of found risks on these trajectories are $R_{AB}=1.1583,R_{AC1B}=R_{AC_{2}B}=0.642,R_{AC_{3}B}=R_{AC_{4}B}=0.93$. Thus, the gain in risk on two-link trajectory over straight path AB is approximately 80%.

### 7.5. Example 5. Independence of Optimal Angle Values $\beta_{1}$* and $\beta_{2}$* from $\beta_{0}$

The last example illustrates Theorem 4. A number of trajectories considered with different end points $B_{i}$, i=1…7 and the same start point *A* for a mobile vehicle with fixed radiation pattern. Optimal two-link paths found using numerical algorithm and shown on the plane $\left(\rho -\rho_{0},\varphi -\varphi_{0}\right)$ in Figure 19. It is clear that all of them consist of line segments of two angles $\beta_{1}$* and $\beta_{2}$*-the optimal angles described in Theorem 4.

## 8. Conclusions

The article considers an optimal trajectory planning problem in a threat environment for any type of a mobile vehicle. The main feature of the problem setting is the non-uniform radiation pattern of the vehicle. The TP-problem is set as a variation’s calculus problem, and then it is solved analytically. A series of lemmas and theorems is formulated and proved in the paper. Necessary and sufficient optimality conditions for trajectory considered and examined. Moreover, specific cases of radiation patterns, degenerating these conditions, are studied. As well as simple one-link optimal trajectories, more complex two-link ones are investigated. All analytical solutions are verified numerically and visualised in Matlab programming modules.

The considered problem and the whole class of such TP problems are extremely relevant to modern-warfare applications. Discussed mobile vehicle can be actualized in practice as a UUV or UAV, for example. Obtained solutions can be helpful for design of on-board real-time algorithms for route planning modules of these objects and modules themselves. As shown in examples solving TP problem can significantly reduce the risk value and provide more safety for UUV on the found optimal route in the threat environment.

Future work can be focused on complication and generalization of the mathematical model and optimization functional of the problem. Current article studies search system, that contains only one detection sensor. Increasing the number of sensors and their processed frequencies presents an interesting and important challenge for applications. Physical features of the signal carrying medium, such as attenuation factor, are also yet to be considered.

## Figures and Tables

**Figure 1 sensors-21-00396-f001:**
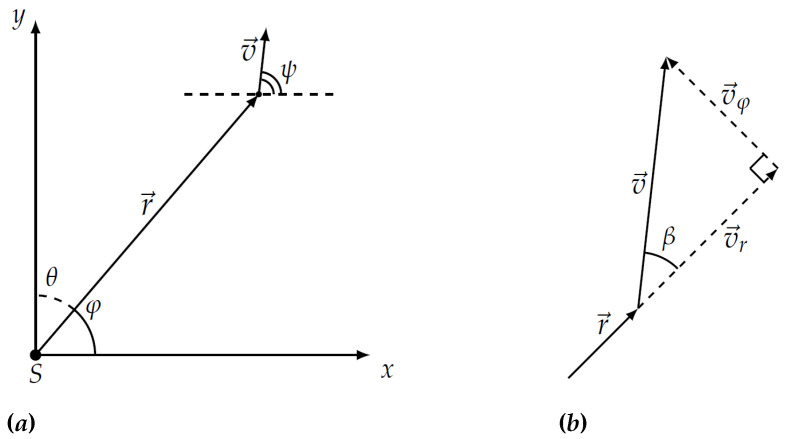
The mobile vehicle in the Cartesian plane with sensor *S* (**a**) and its velocity vectors $\vec{v}$, ${\vec{v}}_{r}$, and ${\vec{v}}_{\varphi }$ (**b**).

**Figure 2 sensors-21-00396-f002:**
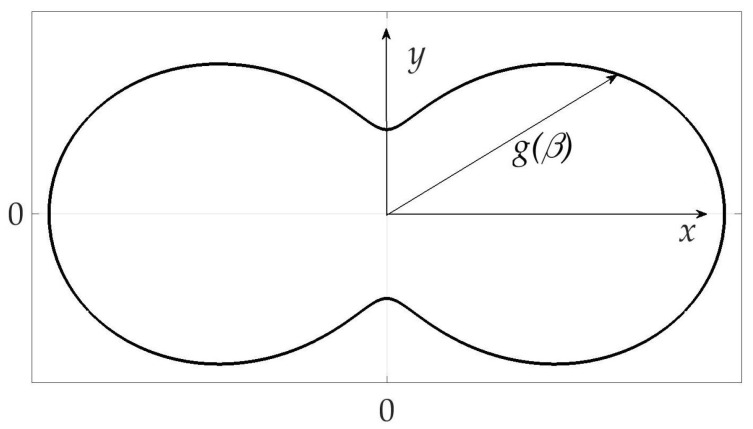
Radiation pattern of the mobile vehicle.

**Figure 3 sensors-21-00396-f003:**
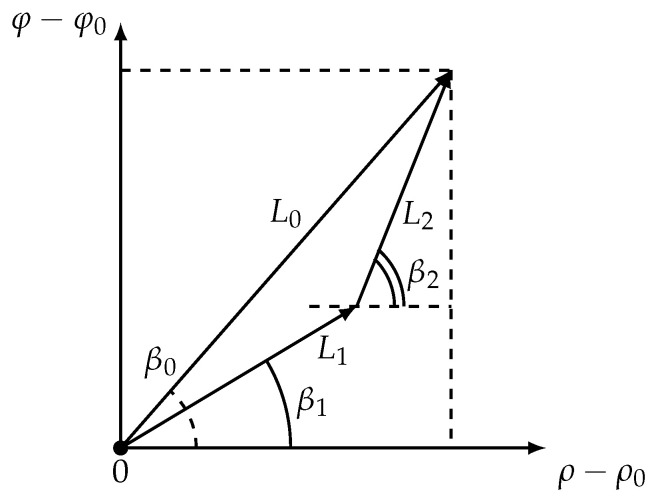
The mobile vehicle in the (ρ,φ) coordinate system.

**Figure 4 sensors-21-00396-f004:**
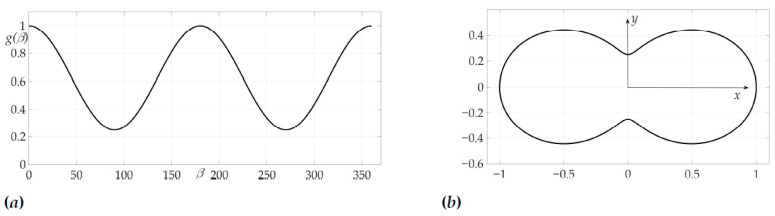
Radiation pattern as an dependence g(β) in polar coordinates (**a**) and on the Cartesian plane (**b**).

**Figure 5 sensors-21-00396-f005:**
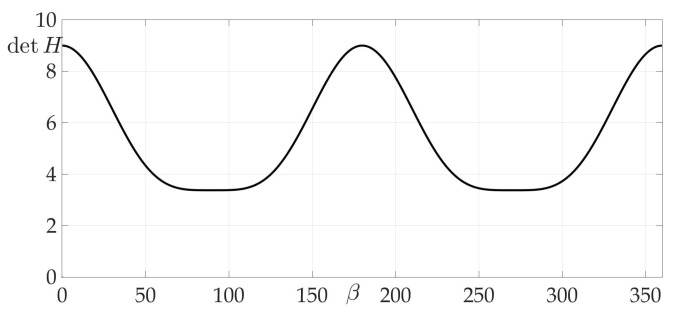
Dependence of Hessian detH from β.

**Figure 6 sensors-21-00396-f006:**
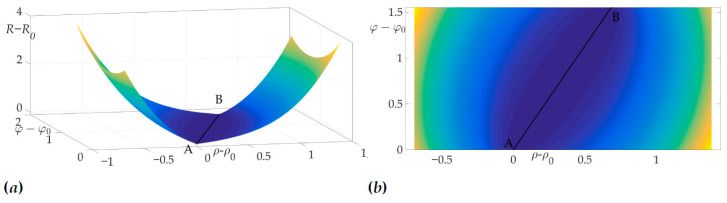
Optimal trajectory in $\left(\rho -\rho_{0},\;\varphi -\varphi_{0},R-R_{0}\right)$ space (**a**) and its projection on $\left(\rho -\rho_{0},\;\varphi -\varphi_{0}\right)$ plane (**b**).

**Figure 7 sensors-21-00396-f007:**
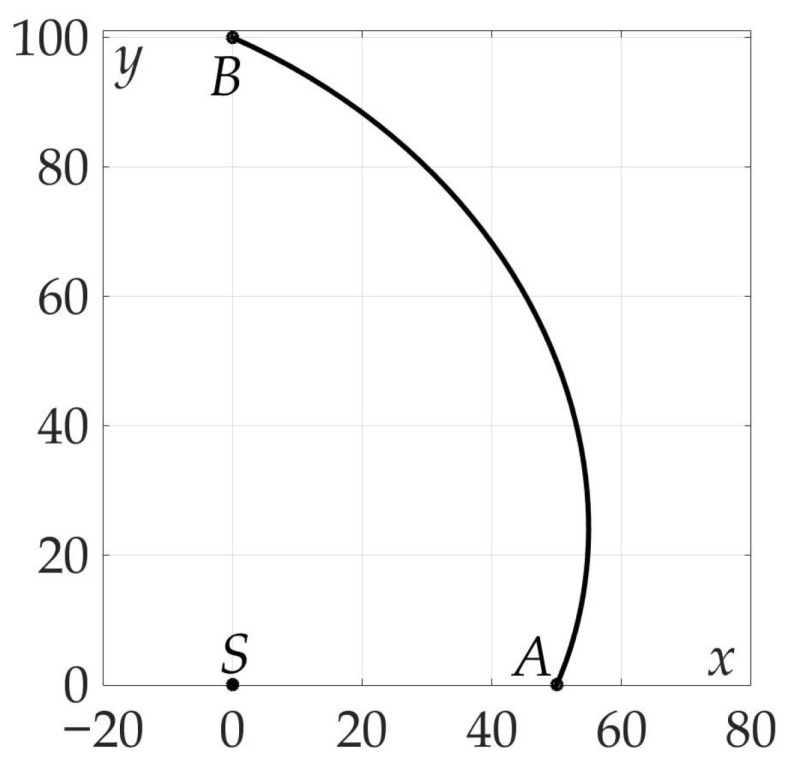
Optimal trajectory on Cartesian plane.

**Figure 8 sensors-21-00396-f008:**
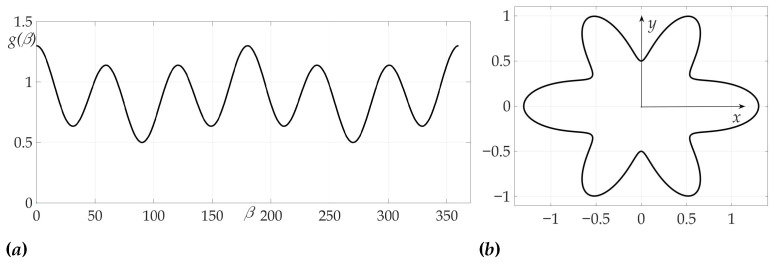
Radiation pattern as an dependence g(β) polar coordinates (**a**) and on the Cartesian plane (**b**).

**Figure 9 sensors-21-00396-f009:**
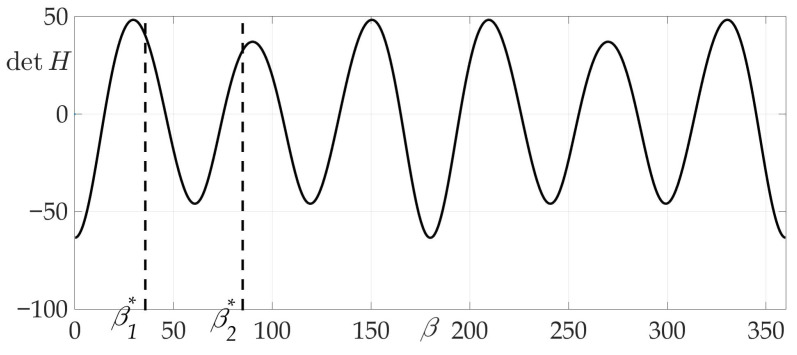
Dependence of Hessian detH from β.

**Figure 10 sensors-21-00396-f010:**
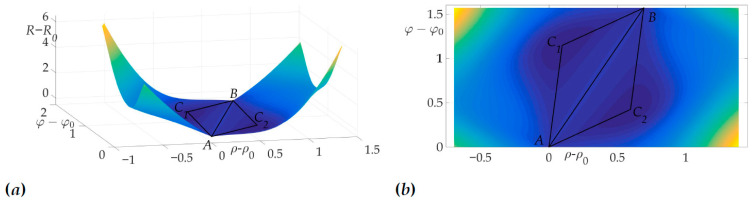
Optimal trajectory in $\left(\rho -\rho_{0},\;\varphi -\varphi_{0},R-R_{0}\right)$ space (**a**) and its projection on $\left(\rho -\rho_{0},\;\varphi -\varphi_{0}\right)$ plane (**b**).

**Figure 11 sensors-21-00396-f011:**
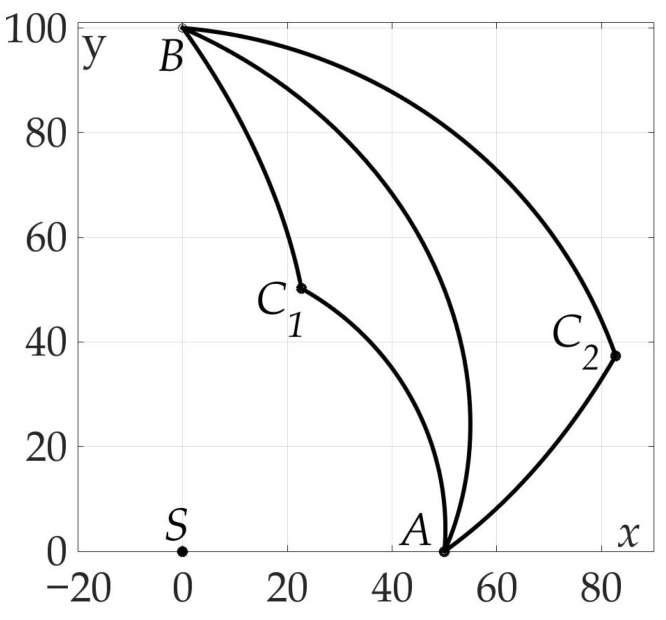
Optimal trajectory on Cartesian plane.

**Figure 12 sensors-21-00396-f012:**
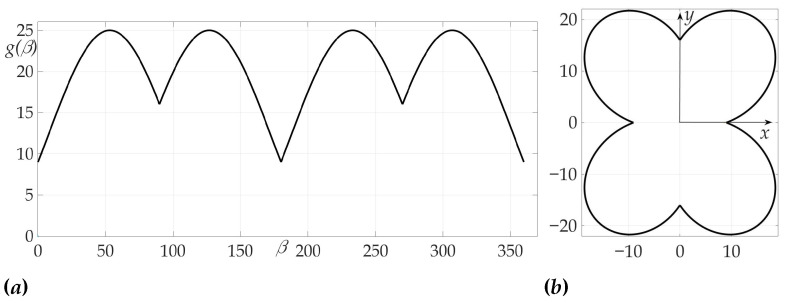
Radiation pattern as an dependence g(β) in polar coordinates (**a**) and on the Cartesian plane (**b**).

**Figure 13 sensors-21-00396-f013:**
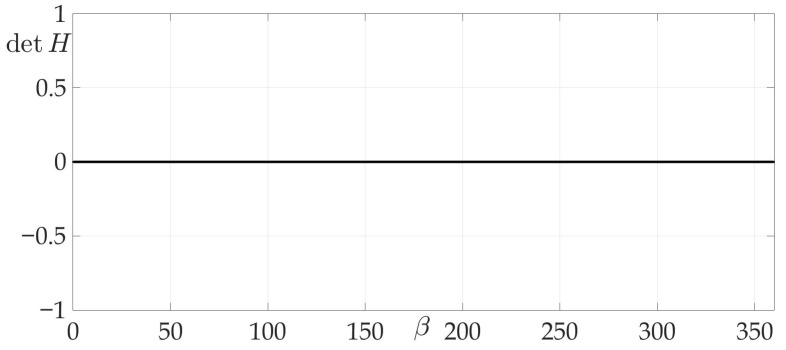
Dependence of Hessian detH from β.

**Figure 14 sensors-21-00396-f014:**
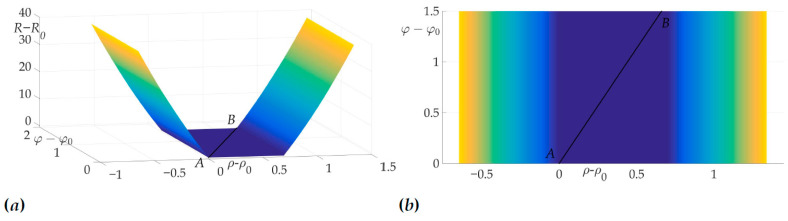
Optimal trajectory in $\left(\rho -\rho_{0},\;\varphi -\varphi_{0},R-R_{0}\right)$ space (**a**) and its projection on $\left(\rho -\rho_{0},\;\varphi -\varphi_{0}\right)$ plane (**b**).

**Figure 15 sensors-21-00396-f015:**
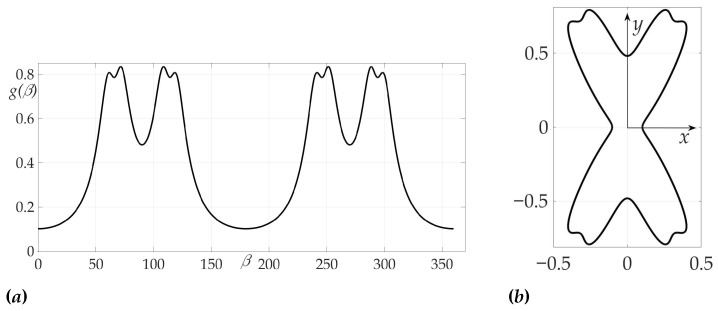
Radiation pattern as an dependence g(β) in polar coordinates (**a**) and on the Cartesian plane (**b**).

**Figure 16 sensors-21-00396-f016:**
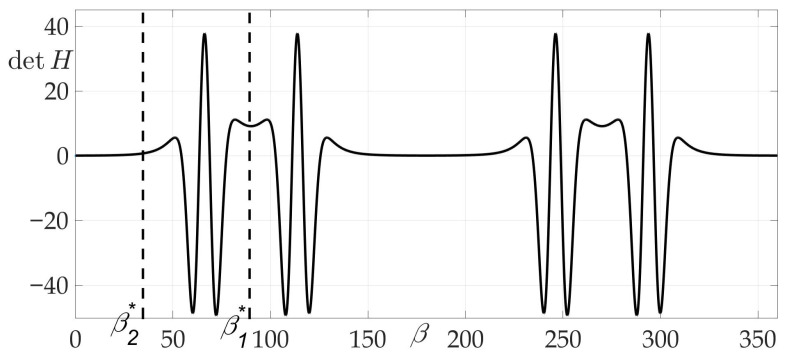
Dependence of Hessian detH from β.

**Figure 17 sensors-21-00396-f017:**
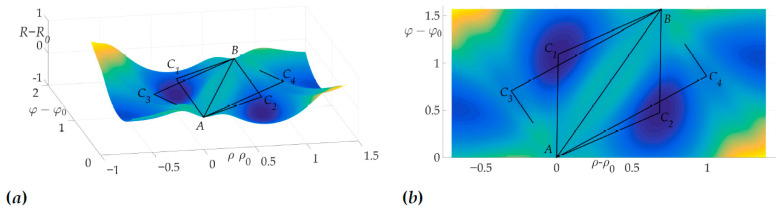
Optimal trajectory in $\left(\rho -\rho_{0},\;\varphi -\varphi_{0},R-R_{0}\right)$ space (**a**) and its projection on $\left(\rho -\rho_{0},\;\varphi -\varphi_{0}\right)$ plane (**b**).

**Figure 18 sensors-21-00396-f018:**
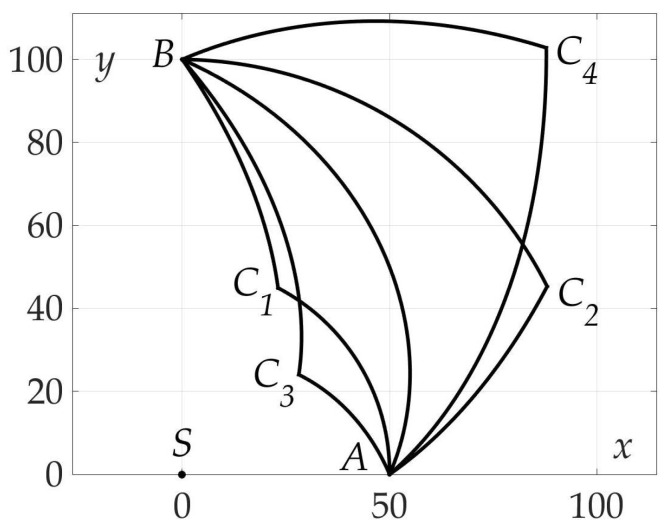
Optimal trajectory on Cartesian plane.

**Figure 19 sensors-21-00396-f019:**
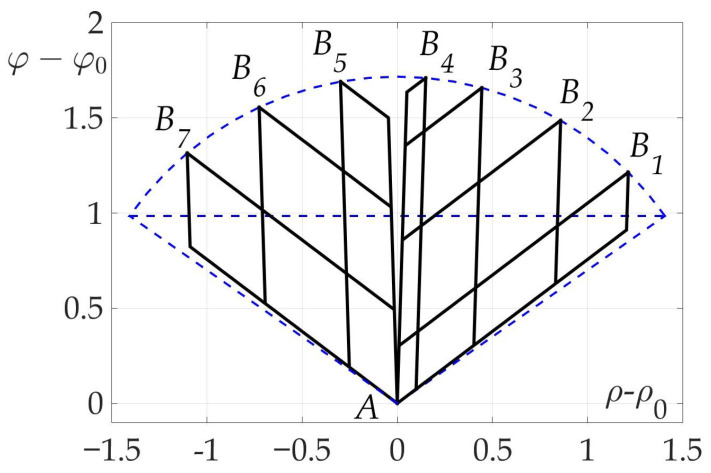
Some optimal two-link paths with different $\beta_{0}$.

## Data Availability

Data sharing not applicable.

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
