# Peer review of "2D Optimal Trajectory Planning Problem in Threat Environment for UUV with Non-Uniform Radiation Pattern"

_sensors, 2021, doi:10.3390/s21020396_

Round 1

Reviewer 1 Report

The article is about an optimal trajectory planning problem of a mobile vehicle which is moving in a threat environment. Mathematical model is impressive and the theory is highly mathematized and very interesting but I couldn't understood the optimization problem itself ! A trajectory (in plane or space) means an infinity points among there are some special ones: a starting point A and a target point B (as are denoted in the examples). As for the optimization problem itself if I can guess the objective function then the constrains are not explicitly specified as is formulate in a regular continuous optimization problem. Please formulate the optimization problem as is explained in the abstract of the paper.

How are the two-link trajectories ? In theory of mechanisms link means a rigid body. Maybe the word "section" or "segment" are better in this case ... Anyway a trajectory is a path in Cartesian space. So is a little strange to represent in Fig. 3 the polar coordinates (radius, angle) in the same manner as is Cartesian planar system (with 2 perpendicular axes) because units are different: meters and radians.

Examples are very interesting and they are according to the theory but is missing a practical example ! Authors should point out at least a practical example with some practical data if possible.

Author Response

Thank you very much for your revision. The answer contains in the attached file.

Reviewer 2 Report

This paper presents a study of the optimal trajectory planning problem in threat environment for UUV.

The main problem I find is that the article seems to be quite similar to their previous one "Path Planning in Threat Environment for UUV with Non-Uniform Radiation Pattern". The same problem with different conditions. Do the new conditions justify another paper? The authors should discuss this fact in detail in the article.

Author Response

(The authors gave the same response as above.)

Reviewer 3 Report

Dear Authors

       Your article "2D Optimal Trajectory Planning Problem in Threat Environment for UUV with Non-Uniform Radiation Pattern". Is quite interesting. Although the reading is quite dense, you do offer conditions for the reader to understand the mathematical development presented. About the text, I have pointed out some suggestions for improvements in the comments in the attached pdf file. My main suggestions:

    1) Highlight in Chapter 1 the similarity of this article with the other work: "Path Planning in Threat Environment for UUV with Non-Uniform Radiation Pattern ". In the current article, you have optimality conditions and refining these conditions, plus a lot of modeling examples has been done to illustrate the finding of two-link and multi-link optimal trajectories. In this article, you managed to solve many theoretical and algorithmic problems for various types of physical fields and signals in the problem of planning the optimal UUV routes that were not properly explored in the first article

    2) Regarding the design of the article, I believe that you will be able to improve the position of the figures as they are previously mentioned in the text and even before the discussions.

    3) On line 170, could you comment a little on the determination of the constants?

     4) In line 175, you should better justify this part: "If sufficient conditions are not fulfilled the optimal trajectory has a more complicated form and consists of several logarithmic spirals".

     5) It would be possible to make available the MatLab scripts that were used in chapter 7. I loved the results and this can facilitate future works.

     I conclude by congratulating them for all the development and writing of an article with a strong theoretical appeal but all the mathematical development presented.

     Happy and blessed Christmas to you!

Author Response

Thank you very much for your thorough revision and your suggestions.
It is always very pleasant to see that reviewer has fully examined the article
and has presented an adequate critique and helpful comments. The answer contains in the attached file. Also moderate English changes have been made according to reviewer’s comments in his pdf review file, which authors are extremely grateful for.
We wish you a Merry Christmas and all the best!

Round 2

Reviewer 1 Report

I understood your point of view and your approach now. As for the word "link" it is maybe used by many specialists from many areas and I am not a native English speaker but in my mind it is linked (!) much more to an existing object then an imaginary "thing" ... As for practical example I think that is better at least to write shortly about one of the practical application. And I will looking forward to seeing your future study with numerical examples which are maybe much more interesting to me !

Author Response

Thank you very much for your revision.

The words about the practical application of current research have been added to Chapter 7 on line 251.

"Current article develops the analytical methods for the constructing of optimal reference trajectories. However, in the first part of work \cite{Sensors} an explicit form criterion for the PP problem for various noise conditions and two types of decision rules are given, based on a comparison of threshold statistics. The real-world data describing this statistics can be used for practical applications of research introduced in these articles. Detection probabilities can be obtained through the calculation of the risk criterion. Certainly, practical aspects make all the analytical models and estimations much more complicated, but still the presented in the article research can be helpful for finding reference paths for UUVs."

Reviewer 2 Report

After reading the explanations, I do not feel that the novelty of the article is enough to be published in Sensors.

Author Response

Thank you very much for your revision.

The words about the practical application of current research have been added to Chapter 7 on line 251.

"Current article develops the analytical methods for the constructing of optimal reference trajectories. However, in the first part of work \cite{Sensors} an explicit form criterion for the PP problem for various noise conditions and two types of decision rules are given, based on a comparison of threshold statistics. The real-world data describing this statistics can be used for practical applications of research introduced in these articles. Detection probabilities can be obtained through the calculation of the risk criterion. Certainly, practical aspects make all the analytical models and estimations much more complicated, but still the presented in the article research can be helpful for finding reference paths for UUVs."

This manuscript is a resubmission of an earlier submission. The following is a list of the peer review reports and author responses from that submission.